# Targeted Therapies and Developing Precision Medicine in Gastric Cancer

**DOI:** 10.3390/cancers15123248

**Published:** 2023-06-19

**Authors:** Rille Pihlak, Caroline Fong, Naureen Starling

**Affiliations:** Gastrointestinal/Lymphoma Unit, The Royal Marsden NHS Foundation Trust, London SW3 6JJ, UK; rille.pihlak@nhs.net (R.P.); caroline.fong@rmh.nhs.uk (C.F.)

**Keywords:** gastric cancer, immunotherapy, HER2, FGFR, GLDN18.2, precision medicine

## Abstract

**Simple Summary:**

Stomach cancers remain highly aggressive cancers with poor patient outcomes; however, in the last two decades there has been a lot of clinical research into understanding targets and potential for targeted treatments in advanced stomach cancers, with some treatments that have already reached patients and have been shown to improve outcomes. In this article, we will summarise the recent advancements in targeted therapies and precision medicine in gastric cancer and discuss new treatments potentially on the horizon.

**Abstract:**

Gastric cancer is an aggressive disease with survival remaining poor in the advanced setting. More than a decade after the first targeted treatment was approved, still only HER2, MSI and PDL-1 status have reached everyday practice in terms of guiding treatment options for these patients. However, various new targets and novel treatments have recently been investigated and have shown promise in improving survival outcomes. In this review, we will summarise previous and currently ongoing studies on predictive biomarkers, possible new targeted treatments, potential reasons for conflicting trial results and hope for the future of precision medicine in gastric cancer.

## 1. Introduction

Gastric and gastroesophageal junction (G/GOJ) cancers are the 17th most common cancers in the UK, with their incidence decreasing in the last two decades [1], although in the advanced setting, survival length is still dismal. Established molecular biomarkers for targeted therapies are uncommon in G/GOJ cancers with only human epidermal growth factor receptor 2 (HER2), microsatellite instable (MSI) and most recently PDL-1 status directing therapeutic options for patients. However, similar to other gastrointestinal cancers, various molecular alterations are currently being investigated, with some early promising results. In this review, we will give an overview of various molecular alterations and predictive biomarkers found in advanced gastric cancer (Figure 1) and potential targeted therapies that could be utilised.

## 2. Immune Checkpoint Inhibitors

Immune checkpoint inhibitors (ICIs) targeting the programmed death-1 (PD-1) and programmed death ligand-1 (anti-PD-L1) pathways have been extensively assessed in multiple tumour types including gastric cancers. Their role in gastric cancers has been investigated with varying degrees of success (Table 1).

### 2.1. PD-L1 Positivity

The anti-PD1 antibody nivolumab was the first ICI to demonstrate efficacy in advanced gastro-oesophageal cancers in the phase III ATTRACTION-2 study, which randomised a chemorefractory, biomarker-unselected Asian population to nivolumab or placebo. The trial met its primary endpoint and demonstrated superior overall survival (OS) in the nivolumab group compared to those who received placebo (mOS: 5.26 months (95% CI 4.6–6.37) versus 4.14 months (95% CI 3.42–4.86); HR 0.63, *p* < 0.001) [12]. Exploratory analysis of PD-L1 tumour cell expression did not show a discriminatory effect on survival. ATTRACTION-2 led to the first approval of ICIs in the chemorefractory setting in Japan, South Korea, Taiwan and Singapore. In a global population, KEYNOTE-059 (phase II) demonstrated an ORR of 11.6% with third-line pembrolizumab, with an improvement in ORR to 15.5% in patients with PD-L1-positive disease, defined as the combine positive score (CPS) of ≥1 [13]. This data formed the basis of a U.S. Food and Drug Administration (FDA) approval for pembrolizumab in this setting for PD-L1-positive disease; this has since been revoked in 2021 in light of therapeutic developments in the first-line setting, as later discussed.

However, pembrolizumab did not significantly prolong PFS and OS against second-line paclitaxel in PD-L1 CPS ≥ 1 patients in the KEYNOTE-061 study and was non-inferior to first-line cisplatin and 5-fluorouracil [14,15]. In both studies, crossing of the pembrolizumab and chemotherapy arms implied an early attrition with ICI monotherapy, but suggested that a subset of patients obtained durable responses from pembrolizumab. Post hoc analyses of these studies have since shown that microsatellite instable (MSI-H), high PD-L1 expressors with CPS ≥ 10 and tumour mutational burden (TMB, defined as ≥10 mutations/Mb) can identify patients most likely to benefit from pembrolizumab and indicate that these parameters could facilitate patient selection for ICIs [16]. In the first-line maintenance setting, two studies failed to show a PFS advantage with ICIs upon the establishment of disease control with induction chemotherapy in PD-L1-unselected patient cohorts: the phase III JAVELIN study which randomised patients to avelumab or chemotherapy/best supportive care, and the phase II PLATFORM study where patients with durvalumab were compared to active surveillance [17,18].

Since then, further studies have demonstrated that ICIs are best utilised in combination with chemotherapy in earlier lines of treatment as the chemotherapy component mitigates the relatively shorter PFS seen with ICI monotherapy. CheckMate 649 was a global study assessing nivolumab in addition to oxaliplatin and fluoropyrimidine chemotherapy as first-line treatment for HER2-negative oesophageal, gastro-oesophageal and gastric adenocarcinoma. In the primary population consisting of patients with a PD-L1 CPS of ≥5, the median OS was 14.4 months with nivolumab and chemotherapy, in comparison with 11.1 months in the chemotherapy arm (HR 0.71 [98·4% CI 0.59–0.86]; *p* < 0.0001) [2]. Hierarchical statistical analysis in the CPS ≥ 1 population and in all patients showed sustained statistical significance in survival benefit (HR 0.71 and HR 0.80 respectively). Concurrently, the ATTRACTION-4 study conducted in an Asian cohort of patients reported an improvement in PFS, but not OS with first-line nivolumab [19]. In comparison to CheckMate 649, the higher proportion of patients who received subsequent lines of systemic therapy in ATTRACTION-4 (39% vs. 66%) may have mitigated an OS difference in the latter study. The third study demonstrating that chemo-immunotherapy is superior to chemotherapy in treatment-naïve advanced gastric adenocarcinoma is the ORIENT-16 study, which reported superior PFS and OS outcomes in a Chinese population with PD-L1 CPS ≥ 5 [20]. These studies have led to chemo-immunotherapy being a new standard-of-care for HER2-negative first-line therapy in advanced gastro-oesophageal adenocarcinoma, although there are geography-dependent nuances in licensing. For instance, nivolumab with chemotherapy is FDA-approved irrespective of PD-L1 whereas the European Medicines Agency approval for this combination limits its use to patients with PD-L1 CPS ≥ 5 only. It should also be noted that the KEYNOTE-062 study also reported that chemo-immunotherapy using pembrolizumab was not superior to chemotherapy alone in patients with PD-L1 CPS ≥ 1 tumours. Interestingly, it is the only first-line chemo-immunotherapy study that used cisplatin within its chemotherapy backbone. Although oxaliplatin may be the better partner for pembrolizumab as it induces immunogenic cell death in vitro, it seems unlikely that the differential platinum component alone would account for these negative results as cisplatin has been successfully combined with chemotherapy in other tumour types. Differences in trial design and statistical considerations may have also contributed to this discrepancy. Most recently, the results from a placebo-controlled global phase III KEYNOTE-859 trial [21] were presented, which randomised 1579 patients to first-line doublet chemotherapy (cisplatin and 5-FU or CAPOX) and placebo or pembrolizumab, and similar to CheckMate 649 showed significant improvement in OS and PFS in the pembrolizumab arm compared to placebo (OS = 12.9 vs. 11.5 and PFS = 6.9 vs. 5.6 months, respectively). Importantly, it also showed no difference between the cisplatin + 5FU and CAPOX chemotherapy backbones [21].

It is recognised that PD-L1 CPS can enrich for patients more likely to respond to ICIs [2,22]. Assessment of PD-L1 CPS itself is associated with a host of intricacies. Briefly, CPS for gastric cancers was developed using the Dako 22C3 assay as a companion diagnostic to pembrolizumab. However, individual ICIs have been developed with their respective companion PD-L1 assays. For example, the Dako 28-8 assay used in CheckMate 649, which currently forms the largest dataset of PD-L1 CPS expression in the advanced setting, reported that 80% and 60% of patients with advanced HER-negative gastro-oesophageal adenocarcinoma have CPS ≥ 1 and ≥ 5 respectively. Based on these data, the CPS can identify a significant proportion of patients who may derive benefit from first-line chemo-immunotherapy. Comparisons of assay concordance have largely been limited to small retrospective studies [23,24], although a head-to-head comparison of both the Dako 28-8 and 22C3 assays derived from 362 gastric cancer tumour samples has shown that the percentage of PD-L1 positivity using the former antibody is twice as high as that of the 22C3 assay at various cut-offs (CPS ≥ 1: 70.3 vs. 49.4%, *p* < 0.001; CPS ≥ 5: 29.1 vs. 13.4%, *p* < 0.001; and CPS ≥ 10: 13.7 vs. 7.0%, *p* = 0.004) [25]. Improving our understanding of PD-L1 assay variation and interchangeability is critical as the current licensing of nivolumab with first-line platinum–fluoropyrimidine chemotherapies in Europe, for example, is restricted to patients with PD-L1 CPS ≥ 5 tumours and assay-independent. Furthermore, it is recognised that PD-L1 expression exhibits marked spatial and temporal heterogeneity, with one study reporting a 61% concordance between primary and metastatic tumours and 63% concordance in PD-L1 expression in paired primary tumour biopsies following exposure to chemotherapy. Although PD-L1 CPS has multiple limitations, a clear advantage is that immunohistochemistry requires a relatively small amount of tissue and can be reported in a timely fashion, two attributes that are highly relevant to successful real-world clinical application.

**Table 1 cancers-15-03248-t001:** Selected ICI clinical trials in advanced oesophagogastric adenocarcinoma.

Chemorefractory							
Trial	Phase	Population	*n*	Trial arms	Response rate	PFS	OS
**ATTRACTION-2** [12]	III	GOJ or gastric; Asian	493	Nivolumab vs. placebo	11% (8–16%) (DCR: 40% vs. 25%)	1.61 vs. 1.45 m (HR 0.6, *p* < 0.0001)	5·26 vs. 4.14 m (HR 0.63, *p* < 0.0001)
**KEYNOTE-059** [16]	II	GOJ or gastric	259	Pembrolizumab	**All patients:** 11.6% **PD-L1 CPS ≥ 1**: 16% **PD-L1 CPS < 1:** 6%	2.0 m	5.5 m
**CheckMate 032** [26]	I/II	Oesophageal adenocarcinoma, GOJ or gastric	160	Nivolumab 3 mg/kg (NIVO3) vs. nivolumab 1 mg/kg + ipilimumab 3 mg/kg (NIVO1 + IPI3) vs. nivolumab 3 mg/kg + ipilimumab 1 mg/kg (NIVO3 + IPI1)	12% vs. 24% vs. 8%	12 months: 8% vs. 17% vs. 10%	12 months: 39% vs. 35% vs. 24%
**JAVELIN Gastric 300** [27]	III	GOJ or gastric	371	Avelumab vs. physician’s choice of weekly paclitaxel or two-weekly irinotecan or BSC	2.2% vs. 4.3%	1.4 vs. 2.7 m	4.6 vs. 5.0 m (HR 1.1)
**Second-line setting**							
Trial	Phase	Population	*n*	Trial arms	Response rate	PFS	OS
**KEYNOTE-061 [14]**	III	Gastric or GOJ	592	Pembrolizumab vs. weekly paclitaxel	**CPS ≥ 1:** 16% vs. 14% **CPS ≥ 10:** 24.5% vs. 9.1% **CPS < 1**: 2% vs. 10.4%	**CPS ≥ 1:** 1.5 vs. 4.1 m (HR 1.27)	**CPS ≥ 1:** 9.1 vs. 8.3 m (HR 0.82; *p* = 0.04) **CPS ≥ 10:** 10.4 vs. 8.0 m (HR 0.64)
**KEYNOTE-181 [28]**	III	Oesophageal adenocarcinoma and SCC	628 *n* = 227 ACC	Pembrolizumab vs. physician’s choice of taxane or irinotecan	-	-	**ITT:** 7.1 vs. 7.1 m (HR 0.89; *p* = 0.056) **PD-L1 CPS ≥ 10**: 9.3 vs. 6.7 m (HR 0.69, *p* = 0.0074) **ACC:** 6.3 vs. 6.9 m
**Maintenance post first-line therapy**							
**JAVELIN Gastric 100** [17]	III	GOJ or gastric	499	Avelumab vs. continuation of chemotherapy or BSC	13.3% vs. 14.4%	3.2 vs. 4.4 m (HR 1.04)	10.4 vs. 10.9 m (HR 0.91; *p* = 0.18)
**First-line setting**							
Trial	Phase	Population	*n*	Trial arms	Response rate	PFS	OS
**KEYNOTE-062 [15]**	III	GOJ or gastric PD-L1 CPS ≥ 1	763	Pembrolizumab vs. chemotherapy + pembrolizumab vs. chemotherapy+ placebo	**P vs. CTx** **CPS ≥ 1:** 14.8% vs. 37.2% **CPS ≥ 10:** 25.0% vs. 37.8%	**P vs. CTx ** **CPS ≥ 1:** 2.0 vs. 6.4 m (HR 1.66) **CPS ≥ 10:** 2.9 vs. 6.1 m (HR 1.10)	**P vs. CTx** **CPS ≥ 1**: 10.6 vs. 11.1 m (HR 0.91; *p* = 0.162) **CPS ≥ 10:** 17.4 vs. 10.8 m (HR 0.69)
					**P vs. CTx + P** **CPS ≥ 1:** 48.6 % vs. 37.2%	**P vs. CTX + P** **CPS ≥ 1:** 6.9 vs. 6.4 m (HR 0.84; *p* = 0.039) **CPS ≥ 10:** 5.7 vs. 6.1 m (HR 0.73)	**P vs. CTX + P** **CPS ≥ 1:** 12.5 vs. 11.1 m (HR 0.85; *p* = 0.046) **CPS ≥ 10:** 12.3 vs. 10.8 m (HR 0.85; *p* = 0.158)
**CheckMate 649 [2]**	III	Oesophageal adenocarcinoma, GOJ and gastric	1581	Nivolumab + chemotherapy vs. chemotherapy	**CPS ≥ 5:** 60% vs. 45%	**CPS ≥ 5:** 7.7 vs.6.0 m (HR 0.68, *p* = 0.0001) **CPS ≥ 1:** 7.5 vs. 6.9 m (HR 0.74) **All patients:** 7.7 vs. 6.9 m (HR 0.77)	**CPS ≥ 5:** 14.4 vs. 11.1 m (HR 0.91, *p* < 0.0001) **CPS ≥ 1:** 14.0 vs. 11.3 (HR 0.77, *p* = 0.0001) **All patients:** 13.8 vs. 11.6 (HR 0.80, *p* = 0.0002)
**ATTRACTION-4 [19]**	II/III	GOJ or gastric; Asian	724	Nivolumab + chemotherapy vs. chemotherapy + placebo	57.5% vs. 47.8%	10.45 vs. 8.34 m (HR 0.68; *p* = 0.0007)	17.45 vs. 17.15 m (HR 0.90; *p* = 0.257)
**KEYNOTE-590 [29]**	III	Oesophageal adenocarcinoma and SCC or Type 1 GOJ (n = 201 oesophageal ACC and Type 1 GOJ)	749	Pembrolizumab + chemotherapy vs. chemotherapy + placebo	**All patients**: 45% vs. 29.3%	**SCC:** 6.3 vs. 5.8 m (HR 0.65; *p* < 0.0001) **CPS ≥ 10:** 7.5 vs. 5.5 m (HR 0.51; *p* < 0.0001) **All patients:** 6.3 vs. 5.8 m (HR 0.65; *p* < 0.0001)	**SCC CPS ≥ 10:** 13.9 vs. 8.8 m (HR 0.57; *p* < 0.0001) **SCC:** 12.6 vs. 9.8 m (HR 0.72; *p* = 0.0006) **CPS ≥ 10:** 13.5 vs. 9.4 m (HR 0.62, *p* < 0.0001) **All patients:** 12.4 vs. 9.8 m (HR 0.73; *p* < 0.0001)
**ORIENT-16 [20]**	III	GOJ or gastric; Asian	650	Chemotherapy + sintilimab vs. chemotherapy + placebo	**All patients:** 58.2% vs. 48.4%	**CPS ≥ 5:** 7.7 vs. 5.8 m (HR 0.628; *p* = 0.0002) **All patients:** 7.1 vs. 5.7 m (HR 0.636; *p* < 0.0001)	**CPS ≥ 5:** 18.4 vs. 12.9 m (HR 0.66; *p* = 0.0023) **All patients:** 15.2 vs. 12.3 m (HR 0.766; *p* = 0.009)
**KEYNOTE-859** [21]	III	GOJ or gastric	1579	Pembrolizumab + chemotherapy vs. chemotherapy + placebo (cisplatin and 5-FU or CAPOX)	**All patients:** 51.3% vs. 42.0%	**All patients:** 6.9 m vs. 5.6 m (HR 0.76; *p* < 0.0001)	**All patients:** OS = 12.9 m vs. 11.5 m (HR 0.78; *p* < 0.0001)

ACC—adenocarcinoma; CI—confidence interval; CPS—combined positivity score; CTx—chemotherapy; HR—hazard ratio; m—months; OS—overall survival; P—placebo; PFS—progression-free survival; SCC—squamous cell carcinoma.

### 2.2. MMR Deficiency

It is clear that the use of ICIs in gastric cancers has its nuances and that predictive biomarkers are crucial to informing patient selection. The most established biomarker of response to ICI in gastric cancers is mismatch repair deficiency (MMRd) or MSI-H, which is a tumour- and histology-agnostic biomarker of ICI efficacy. Proficient cellular mismatch repair (MMR) machinery detects and replaces single-nucleotide mismatches and corrects small insertions and deletions that occur during DNA replication, particularly at repetitive sequences known as microsatellites. Loss of MMR integrity (MMRd) therefore leads to MSI-H, which is characterised by large numbers of frameshift and single-nucleotide variants resulting in a high tumour mutational burden (TMB) [30]. In gastric cancer, MSI-H/MMRd disease occurs with a relatively low prevalence of approximately 8% and 4–5% in patients with surgically resectable and advanced gastro-oesophageal disease, respectively [31,32]. A recently published meta-analysis of 17 randomised controlled phase III trials with a pooled dataset of 11,166 patients with oesophagogastric cancers reported MSI-high as the strongest predictor of ICI benefit in patients with adenocarcinoma (*n* = 6099) treated with immunotherapy or chemo-immunotherapy regimens (HR 0.35, 95% CI 0.22–0.55 for MSI-H vs. HR 0.82, 95% CI 0.74–0.91 for MSI-low) followed by PD-L1 CPS (HR 0.73, 95% CI 0.66–0.81 for CPS high vs. HR 0.95, 95% CI 0.84–1.07 for CPS low) [33]. This echoes findings from a smaller meta-analysis of 2545 patients from four phase III randomised controlled trials (KEYNOTE-062, CheckMate 649, JAVELIN Gastric 100 and KEYNOTE-061) investigating treatment with or without anti-PD-1 agents in advanced gastric cancer, which reported an HR OS benefit of 0.34 (95% CI: 0.21–0.54) in patients with MSI-H status versus 0.85 (95% CI: 0.71–1.00) for microsatellite stable patients. The odds ratio for response was 1.76 (95% CI 1.10–2.83; *p* = 0.02) for MSI-H patients [32]. These results indicate that patients with MSI-H gastric cancer are a highly immunosensitive, albeit small, population. However, the presence of MSI-H/MMRd disease is not tantamount to ICI efficacy as only 50% of patients with MSI-H tumours have an objective response to ICIs, suggesting a degree of intrinsic resistance to these agents [34]. Data derived from a study of 19 Asian patients with MSI-H gastric cancers treated with second- or third-line pembrolizumab in the context of a phase II study described that tumours with features such as higher TMB, T-cell receptor (TCR) pathway activation and a more diverse TCR repertoire, and the presence of stem-like exhausted CD8+ T-cells are associated with benefit from pembrolizumab. By contrast, tumours with a lower TMB, WNT pathway, CDH1, JAK2, ERBB2 and FGFR2 alterations as well as terminally differentiated exhausted CD8+ T-cells were pembrolizumab-insensitive [34]. Even within MSI-H/MMRd patients, a degree of response heterogeneity to ICIs exists, and further research into predictive biomarkers and therapeutic strategies for these patients is required, with clinical trials currently ongoing to investigate this [35].

### 2.3. EBV Positivity

Another biomarker that can be integrated into routine clinical practice is Epstein–Barr virus (EBV) positivity, which is found within malignant epithelial cells in 9% of gastric cancers [36]. EBV-positive tumours have a distinct molecular profile with high *PD-L1* and *PD-L2* expression, marked intra- or peritumoural immune cell infiltration, extreme DNA hypermethylation and recurrent *PIK3CA* mutations [37]. Although EBV positivity is relatively rare, these patients have been shown to be sensitive to ICI monotherapy. For example, all 6 EBV-positive patients recruited into a single-centre phase II trial of 61 biomarker-unselected patients with chemorefractory gastric cancer in Korea achieved partial responses with a median duration of response of 8.5 months with pembrolizumab [38]. A second study reported a complete response lasting >30 months in an EBV-positive patient who received ICI monotherapy [39]. Although the evidence supporting its predictive role is provocative, the relatively low prevalence of EBV may hamper further efforts in assessing its utility as a predictive biomarker of response to ICI. However, EBV testing should be carried out where available as it has the potential of widening treatment options, particularly for patients who remain fit for later-line therapies where options are scarce.

### 2.4. TMB High

Tumour mutational burden (TMB), the total number of mutations per coding area of a tumour genome, can also predict ICI response. Highly mutated tumours, such as MSI-H tumours, are more likely to harbour neoantigens that can enhance immunogenicity. TMB can be determined via whole exome sequencing or targeted gene panel, and the data suggest good concordance between both methods. Pembrolizumab is FDA-approved for all solid tumours with a TMB of ≥10 mutations/Megabase (Mb) based on data from KEYNOTE-158, a multi-cohort basket study [40]. It should be noted that KEYNOTE-158 did not include a gastric adenocarcinoma cohort. The median TMB in oesophagogastric adenocarcinoma ranges between 5 and 6 mutations/Mb but there is currently no consensus on the threshold that defines ‘high TMB’ in this tumour type. In gastric cancer, a post hoc analysis of patients recruited into KEYNOTE-061 demonstrated a strong association between high TMB, defined as 175 mutations/exome using WES or 10 mutations/Mb using FoundationOne^®^CDx, and improved ORR, PFS and OS in pembrolizumab-treated patients (*p* < 0.0001 and *p* ≤ 0.003 for all outcome measures for WES and FoundationOne^®^CDx, respectively) [41]. These significant associations between high TMB and outcome were not reproduced in patients who were randomised to receive paclitaxel in the study. If next-generation sequencing is accessible, TMB can therefore be considered to select patients for immunotherapy, although caveats surrounding the supporting data outlined above should be considered.

## 3. HER2-Targeted Therapies

HER2 is overexpressed in 15–20% of gastric cancers and is most commonly associated with the intestinal subtype and well- to moderately differentiated adenocarcinomas [5]. As a member of the epidermal growth factor family of receptor tyrosine kinases, HER2 heterodimerisation leads to signalling via the RAS/MAPK and PI3K/Akt pathways to induce tumour cell proliferation, invasion and metastasis [42].

HER2 overexpression is induced by *HER2* gene amplification. In the management of advanced oesophagogastric cancer, it represents the first routinely targeted predictive biomarker with implications for clinical decision-making and patient outcomes (Table 2). This is underpinned by the results from the ToGA study, which demonstrated that the addition of trastuzumab, a monoclonal antibody against HER2, to platinum–fluoropyrimidine chemotherapy led to a significant improvement in median OS over chemotherapy alone (mOS: 13.8 vs. 11.1 months, respectively; HR 0.74; 95% CI 0.60–0.91; and *p* = 0.0046) in patients with HER2 3+ staining on immunohistochemistry or *HER2* gene amplification by fluorescence in-situ hybridisation (FISH) [43]. Although a statistically significant survival benefit was seen in the overall trial population, post hoc subgroup analysis showed that the addition of trastuzumab to chemotherapy led to the most pronounced survival benefit in patients with IHC 2+/FISH positive or IHC 3+ tumours (mOS: 16.0 vs. 11.8 months; HR 0.65 (95% CI 0.51–0.83)). These parameters have since defined ‘HER2-positivity’ in oesophagogastric cancers and determine patient eligibility for trastuzumab and platinum–fluoropyrimidine doublet chemotherapy, the current gold standard first-line approach for advanced HER2-positive oesophagogastric cancers.

Despite the initial success of ToGA, several trials investigating other agents targeted against HER2 have failed to lead to a change in practice. This is in distinct contrast to the treatment landscape of HER2-positive breast cancers, showing that while HER2-positive cancers are a molecularly-defined subset of tumours, contradictory trial results point to distinct tumour biology between both tumour types. HER2 expression determined via immunohistochemistry also has a more heterogenous basolateral membrane staining pattern when compared to breast cancers, leading to the development of a HER2 scoring system specific to gastric cancers [44,45]. Additionally, downregulation of HER2 expression has been demonstrated in up to 30% of paired biopsies following trastuzumab-containing regimens and is associated with a lack of response to second-line trastuzumab emtansine [46]. Acquired resistance to HER2-targeted therapies is also attributed to loss of the trastuzumab binding site on the HER2 receptor secondary to aminotruncation, *FGFR3* amplification and activation of the MAP/ERK and PI3K/mTOR downstream pathway signalling [47,48,49]. An understanding of the primary and secondary resistance mechanisms has led to the development of novel agents that are trying to overcome these. One example is antibody-drug conjugates (ADCs), which comprise a monoclonal antibody conjugated to a cytotoxic payload via a chemical linker [50]. Trastuzumab deruxtecan (T-DXd) is an ADC which is composed of a humanised monoclonal anti-HER2 antibody joined to a topoisomerase I inhibitor, at a drug-to-antibody ratio of 8:1 [51]. Its peptide-based linker is cleaved by intracellular lysosomal enzymes, precipitating the release of its cytotoxic payload upon internalisation into a cancer cell and as its drug payload is also membrane-permeable, it can diffuse into neighbouring cells to facilitate a bystander-killing effect. This unique feature may be particularly valuable in overcoming the barriers posed by heterogenous HER2 expression in gastric cancers.

At present, T-DXd is licensed for HER2-positive advanced oesophagogastric adenocarcinoma in the chemo-refractory setting in Asia and in the trastuzumab-refractory setting in the US. This licensing is based on the DESTINY-Gastric01 study which randomised 187 pre-treated Asian patients to receive T-DXd or investigator’s choice chemotherapy (either irinotecan or paclitaxel) and reported that patients with T-DXd had a significantly higher ORR determined by independent central review of 51% compared to 14% with chemotherapy (*p* < 0.001) [52]. Additionally, T-DXd also led to significantly longer overall survival compared to chemotherapy (median OS: 12.5 vs. 8.4 months; R 0.59; and *p* = 0.001). The most common side effects associated with T-DXd were myelosuppression, anorexia, nausea and diarrhoea, and interstitial lung disease was recorded in 10% of the patients recruited. Although the majority of these were low grade (70%), ILD represents an important risk associated with this drug and should be actively monitored for. Therefore, DESTINY-Gastric01 represents a milestone in the management of HER2-positive oesophagogastric cancers as the first randomised study to demonstrate an improvement in trastuzumab-refractory HER2-positive patients.

The subsequent DESTINY-Gastric02 phase II single-arm study reported an ORR of 38% with second-line T-DXd in a cohort of 79 Western patients who had progressed on trastuzumab-containing regimens during primary analysis [53]. These results point towards a clinical benefit of T-DXd in a Western population, and the phase III DESTINY-Gastric04 trial is currently recruiting patients from a global population to determine if T-DXd can improve OS over paclitaxel and ramucirumab.

Due to its bystander effect, T-DXd also holds therapeutic potential in HER2-low disease [54], which to date has been treated as a HER2-negative disease. Phase II evaluation in exploratory cohorts of the DESTINY-Gastric01 study reported an ORR of 26.3% (*n* = 5/19) and 9.5% (*n* = 2/21) in patients with HER2 IHC 2+/ISH- and IHC 1+ disease, respectively [54]. Therefore, these data provide preliminary evidence for a new paradigm shift which is currently being explored in further detail in larger datasets.

Immunotherapy-based approaches are also being pursued as new avenues in the management of HER2-positive disease. The addition of pembrolizumab to trastuzumab and first-line platinum–fluoropyrimidine in the KEYNOTE-811 trial has yielded interim analysis results and a significantly higher ORR of 74.4% was reported in patients who were randomised to receive pembrolizumab in comparison to 51.9% in those who received placebo (*p* = 0.00006) with standard-of-care therapy [55]. This combination currently holds FDA accelerated approval for treatment-naïve HER2-positive gastric cancers. Further survival results from the KEYNOTE-811 trial are eagerly awaited and if positive, the combination is likely to supersede chemotherapy and trastuzumab as the gold standard first-line therapy. Similarly, the first-line DESTINY-Gastric03 trial is currently recruiting patients and investigating T-DXd in combination with chemotherapy or immunotherapy, which will add further results to this field [56].

Other immunotherapy-based agents that have reached phase III evaluation are margetuximab and zanidatamab. Margetuximab is a bispecific anti-HER2 antibody which has been engineered to bind with increased affinity to both lower and higher affinity forms of CD16A, an Fc-receptor on natural killer cells and macrophages which mediate antibody-dependent cellular cytotoxicity. When assessed in combination with pembrolizumab in 92 patients with HER2-positive gastric or GOJ cancers refractory to trastuzumab, an ORR of 18% and median OS of 12.48 months were observed [57]. Interestingly, the ORR was higher at 44% in patients who were both HER2 IHC3+ and PD-L1-positive. Patients with detectable HER2 amplification on ctDNA also had a higher response rate compared to those with undetectable levels. Margetuximab is also being assessed in combination with another anti-PD1, INCMGA00012 or the anti-LAG inhibitor retifanlimab as first-line therapy in the phase II/III MAHOGANY study. Interim analysis results from the first 40 non-MSI-H patients treated with margetuximab and retifanlimab in Cohort A of MAHOGANY reported an ORR of 53% and a median DoR of 10.3 months [58]. Although concurrent advances in HER2-positive OG adenocarcinoma precipitated the early discontinuation of this cohort, these results indicate that the chemotherapy-free regimen of margetuximab and retifanlimab induces similar rates of radiological responses alongside a more favourable toxicity profile in a biomarker-selected population of HER2- and PD-L1-positive patients compared to chemotherapy with trastuzumab.

Simultaneous targeting of extracellular domains 4 and 2 using trastuzumab and pertuzumab, respectively, did not lead to a survival advantage in a trastuzumab-refractory population in the JACOB trial [59]. However, this approach is being revisited after preclinical studies have shown that concurrent targeting of these epitopes with the bispecific zanidatamab induced higher levels of HER2 inhibition and anti-tumour activity compared to trastuzumab and pertuzumab [60]. A DCR and ORR of 56% and 43%, respectively, were reported in a phase I study of 33 patients with heavily pre-treated HER2-positive solid tumours, 11 of which had oesophagogastric adenocarcinoma [61]. A subsequent phase II study investigating zanidatamab with first-line platinum–fluoropyrimidine chemotherapy reported an ORR of 75% and median DoR of 16.4 months [62]. The clinical evaluation of zanidatamab as first-line therapy in HER2-positive oesophagogastric adenocarcinoma continues in combination with chemo-immunotherapy (NCT04276493) [63].

Other novel HER2-targeted agents being assessed in OG adenocarcinoma are tyrosine kinase inhibitors. Tucatinib is a highly selective small molecule TKI against HER2 with confirmed clinical activity in mouse models derived from N87 gastric cancer cell lines [64]. Tucatinib has been shown to be efficacious in HER2-positive breast and colorectal cancers [65,66], and the phase II/III MONTAINEER-02 trial will assess tucatinib, trastuzumab, ramucirumab and paclitaxel as second-line treatment in HER2-positive gastric cancers (NCT0449924) [67].

**Table 2 cancers-15-03248-t002:** Selected randomised HER2-targeted clinical trials in advanced oesophagogastric adenocarcinoma.

Trial	Phase	N	HER2 Definition	Trial Arms	Results
**First-line therapy**					
ToGA [43]	III	594	IHC 3+ and/or ISH-positive	Capecitabine or 5-FU, cisplatin +/− trastuzumab	mOS: 13.8 vs. 11.1 months (HR 0.74, *p* = 0.0046)
TRIO-013/LOGiC [68]	III	545	IHC3+ and/or ISH-positive	Capecitabine, oxaliplatin +/− lapatinib	mOS: 12.2 vs. 10.5 months (HR 0.91, *p* = 0.32)
JACOB [59]	III	780	IHC 3+ or IHC 2+ ISH-positive	Capecitabine or 5-FU, cisplatin, trastuzumab +/− pertuzumab	mOS: 17.5 vs. 14.2 months (HR 0.84, *p* = 0.057)
HELOISE [69]	IIIb	248	IHC 3+ or IHC 2+ ISH-positive	Cisplatin, capecitabine, trastuzumab 8 mg/kg loading dose + 6 mg/kg or 10 mg/kg maintenance dose	mOS: 12.5 vs. 10.6 months (HR 1.24, *p* = 0.2401)
**Second-line therapy**					
TyTan [70]	III	261	ISH-positive	Paclitaxel +/− lapatinib	mOS: 11.0 vs. 8.9 months (HR 0.84, *p* = 0.10)
GATSBY [71]	II/III	302	IHC 3+ or IHC 2+ ISH-positive	Trastuzumab emtansine vs. taxane	mOS: 7.9 vs. 8.6 months (HR 1.15, *p* = 0.86)
T-ACT [72]	II	91	IHC 3+ or IHC 2+ ISH-positive	Paclitaxel +/− trastuzumab	mPFS: 3.2 vs. 3.7 months (HR 0.91, *p* = 0.33)
**Third-line therapy**					
DESTINY-Gastric01 [54]	II	187	IHC 3+ or IHC 2+ ISH-positive	Trastuzumab deruxtecan vs. physician’s choice chemotherapy (irinotecan or paclitaxel)	ORR: 42.8% vs. 12.3%

CI: confidence interval; HR: hazard ratio; IHC: immunohistochemistry; ISH: in situ hybridisation; ORR: overall response rate; OS: overall survival; PFS: progression-free survival.

## 4. CLDN 18.2

Tight junction molecule claudin-18 isoform 2 (CLDN18.2) is expressed in around 40% of patients with gastric cancers [73,74] and is usually not seen in non-malignant tissues besides gastric mucosa, making it a useful potential target for therapies. Zolbetuximab is a monoclonal antibody against CLDN18.2 and has been investigated in various phase II trials as monotherapy or combined with standard chemotherapy [3,75], most recently in a phase III combination SPOTLIGHT trial, the results of which have been presented [76]. As a single agent in the phase II MONO trial, zolbetuximab ORR was shown to be 9% in CLDN18.2-expressing tumours, and in the subgroup of moderate to strong CLDN18.2 expression (≥70% of tumour cells), the ORR was 14%. The phase II FAST trial investigated zolbetuximab combined with ECX (epirubicin, cisplatin and capecitabine) first-line chemotherapy in patients with CLDN18.2 expression in ≥40% tumour cells and showed an ORR of 39%, compared to 25% in the chemotherapy alone arm [3]. The FAST trial also showed statistically significant improvement in OS with 13.0 vs. 8.3 months in the zolbetuximab arm for the whole trial population. In the moderate-to-strong expression group, defined as CLDN18.2 expression in ≥70% of tumour cells, OS increased further to 16.5 vs. 8.9 months with or without zolbetuximab, respectively [3]. On the other hand, there was no statistically significant OS difference (8.3 vs. 7.4 months, HR 0.78, and *p* = 0.4) in the subgroup of patients with 40–69% of tumour cells positive for CLDN18.2.

Phase III clinical trials are currently ongoing with zolbetuximab or placebo added to standard first-line chemotherapy [77,78] and the results of the first SPOTLIGHT trial have been recently presented [76]. This phase III trial randomised 575 patients to either zolbetuximab or placebo using mFOLFOX6 (5-FU, leucovorin and oxaliplatin combination chemotherapy) as the chemotherapy backbone and included patients with moderate-to-strong CLDN18 staining in ≥75% of tumour cells on IHC. The study results showed a median PFS of 10.61 vs. 8.67 months (HR 0.751 and *p* = 0.0066) and a median OS of 18.23 vs. 15.54 months (HR 0.750 and *p* = 0.0053) with or without zolbetuximab, respectively. The most common treatment-related adverse events (TRAE) were nausea, vomiting and appetite loss in the zolbetuximab arm, and TRAEs lead to discontinuation of the study drug in 13.6% of patients compared to 2% in the placebo group [76]. A median OS of 18.23 months is the longest OS shown in a phase III trial for G/GOJ adenocarcinomas; however, it remains to be seen how quickly these results will lead to regulatory approvals and when the combination will become available for patients around the world. Additionally, future trials will need to investigate what to do in the first-line setting if both CPS and CLDN18.2 are high, although it is currently thought that there might be less overlap between these two biomarkers [79].

In addition to zolbetuximab, there are various other CLDN18.2-targeting agents such as monoclonal and bispecific antibodies, chimeric antigen receptor T (CAR-T) cells and ADCs, currently being investigated in early-phase clinical trials [80].

One of the first results from an interim analysis of CLDN18.2-targeted CAR-T cells (CT041) showed promising results in various previously treated GI cancers [81]. The study included 28 patients with G/GOJ cancers, most of whom had received at least two prior lines of treatment. In these 28 patients, ORR was 57.1%, the disease control rate (DCR) 75.0% and the 6-month OS rate was 81.2%. Around half of these patients had high CLDN18.2 expression (≥70%), around 35% had medium expression (40–69%) and 13% had low expression (≤40%) [81]. Therefore, as seen in HER2-targeted agents, the novel CLDN18.2-targeting agents are also showing some potential results in tumours with low target expression, highlighting that the subgroup of patients that could benefit from these treatments could be wider [80].

## 5. FGFR

Fibroblast growth factor receptor 2 (FGFR2) alterations are found in between 9–61% of patients with gastric cancer [82,83,84] depending on the stage and methods used. Several FGFR inhibitors have been investigated as monotherapy or combined with chemotherapy in patients with FGFR2-overexpressed or FGFR2-amplified gastric cancers with mixed results.

As monotherapy, the pan-FGFR tyrosine kinase inhibitor (TKI) AZD4547 did not show any improvement in progression-free survival (PFS) in the second-line setting of FGFR2-amplified gastric cancers compared to standard-of-care paclitaxel chemotherapy [85]. Biomarker analysis from the study did show marked intratumoural heterogeneity which the authors predicted could have led to the negative results [85,86].

Results from the phase II FIGHT trial with bemarituzumab, a monoclonal antibody selectively binding to FGFR2b, combined with a FOLFOX6 chemotherapy regimen in the first-line setting showed FGFR2b overexpression (by immunohistochemistry) or *FGFR2* gene amplification (by ctDNA) in 30.2% of the pre-screened HER2-negative patients [4]. There was a significant improvement in OS when bemarituzumab was combined with FOLFOX6 compared to placebo: OS 19.2 vs. 13.5 months, respectively. In the subgroup of patients with ≥10% FGFR2b+ (by IHC) the OS was 25.4 months with bemarituzumab compared to 11.1 months with placebo [4]. A phase III FORTITUDE-102 clinical trial investigating the combination of bemarituzumab with first-line chemoimmunotherapy is currently ongoing [87].

Similarly, multiple other FGFR-targeted therapies are currently being investigated in phase I-II trials in this setting [88,89,90].

## 6. Homologous Recombination Deficiency (HRD)

Similar to other cancer types, HRD somatic and germline mutations such as BRCA 1/2, PALB2, ATM, ARID1A and others have been found in around 10–15% of patients with gastric cancer [6,91,92]. These alterations have been found to be predictive of response to Poly (ADP-ribose) polymerase (PARP) inhibitor and platinum chemotherapy treatment in various cancers; however, the role of HRD mutations in gastric cancer is still unclear.

Earlier studies found loss of heterozygosity (LOH) high signature as a marker for HRD in around 14% of patients with G/GOJ cancers and showed potential benefit when treating these patients with platinum agents [93]. However, in the phase III GOLD trial, adding PARP inhibitor olaparib to second-line paclitaxel treatment in patients with advanced gastric cancer showed no statistically significant improvement in OS in the whole study population or in the ATM-negative subgroup [94]. The later presented biomarker analysis of the GOLD study similarly showed no specific subgroups that were benefiting from the addition of olaparib [95]. One potential reason proposed for the negative results of this trial was that the olaparib dose in the study was lower than in previously positive monotherapy trials in other cancers, again highlighting the toxicity issues of combining PARP inhibitors with chemotherapy. Various other clinical trials are currently ongoing investigating olaparib as a single agent or in combination with immunotherapy and targeted therapy [96,97,98].

## 7. Tumour Agnostic Targets

Similar to other cancer types, a small percentage of tumour-agnostic targets have been also found in G/GOJ cancers.

### 7.1. KRAS^G12C^ Mutation

Until recently, KRAS (KRAS proto-oncogene) mutations in cancers were thought to be “undruggable”, but novel small molecule inhibitors have now been developed that can specifically target the KRAS^G12C^ mutations. However, out of all KRAS mutations in GI cancers, only a small percentage is G12C. In a retrospective review of next-generation sequencing (NGS) results of more than 17 000 GI cancers, only 0.6% of gastric cancers and 0.3% of oesophageal [9] cancers harboured this specific KRAS^G12C^ mutation.

In the phase I CodeBreaK100 trial, KRAS^G12C^ inhibitor sotorasib showed an ORR of 7.1% in various GI cancers (*n* = 59) including one oesophageal cancer [99]. Another KRAS^G12C^ inhibitor adagrasib was investigated in the KRYSTAL-1 multicohort phase I/II study and showed an ORR of 35% in the non-pancreatic GI cancers group. Phase II trials with both inhibitors are currently ongoing as a single agent or in combination with other anti-cancer treatments.

### 7.2. BRAF^V600E^

BRAF-B-Raf proto-oncogene (BRAF) V600E mutation has been found in 0.4% of patients with G/GOJ cancers [10]. In a phase II basket trial with various BRAF^V600E^ mutant non-melanoma cancers (including oesophageal), the BRAF inhibitor vemurafenib showed an ORR of 33% and a median duration of response of 13 months [100]. Due to the low prevalence of BRAF^V600E^ mutations in G/GOJ cancers, it is not known if these results translate to improved outcomes for G/GOJ cancers. In BRAF^V600E^-mutated colorectal cancers, the combination treatment of BRAF inhibitor encorafenib and anti-EGFR monoclonal antibody cetuximab has shown positive results in a phase III BEACON trial and became the standard second-/third-line treatment for these cancers [101]. It has also been demonstrated that in lower GI cancers BRAF inhibitors alone showed limited results, but combined with anti-EGFR monoclonal antibodies lead to increased antitumour activity. It is hypothesised that this might be similar in other GI cancers and current trials are investigating various combinations.

### 7.3. NTRK Fusion

Neurotrophic receptor tyrosine kinase (NTRK) fusions have been found in <1% of patients with G/GOJ cancers [11]. Various tropomyosin receptor kinase (TRK) inhibitors have been investigated in basket trials combining various cancers with NTRK gene fusions. In the combined data of three ongoing phase I/II clinical trials, the TRK inhibitor entrectinib showed an ORR of 57% and a median duration of response of 10 months [11]. Similarly, in a combination of three phase I-II trials, another TRK inhibitor larotrectinib reached an ORR of 75%, with the median duration of response not reached [102].

## 8. Challenges for Precision Medicine in Gastric Cancer

### 8.1. Heterogeneity of Targets in Gastric Cancers

One of the main challenges for targeted therapies in this patient group has been the heterogeneity of targetable alterations. Both spatial and temporal molecular heterogeneity of targets have been seen in many studies, makes running and understanding clinical trials more complex, as these may be the cause of failure of targeted treatments and lead to negative trial results.

For example, Pectasides et al. showed significant genomic heterogeneity between primary and metastatic lesions in G/GOJ cancers, where in around 42% of the cases, single nucleotide and insertion/deletion mutations differed at baseline [103]. Similar discordance was seen within surgical primary cancer and between resected primary cancer and recurrent cancer metastasis [103]. High intratumoural heterogeneity has also been found in other studies, in around 46% of patient resection samples, and has been linked to potentially worse survival outcomes [104].

The phase II platform PANGEA trial investigated sequential doublet chemotherapy combined with individually matched targeted treatment at baseline and serially over three lines of treatment [105]. During the study, patients had their tumour samples taken at baseline and at each progression to assign them to a biomarker subgroup. The trial showed that at progression after first-line treatment, 49% of patients were assigned to a different biomarker group compared to baseline, and after second-line treatment, a further 48% changed the assigned treatment group. This highlights biomarker heterogeneity sequentially over each treatment line due to either loss of the biological target or other acquired resistance alterations. These results also showed that in around a third of patients (35%), the metastasis molecular profile differed from the primary cancer one, leading to its being assigned to a different targeted therapy group. The PANGEA trial also succeeded in showing improved outcomes of personalized treatment strategies compared to historical controls at baseline through multiple lines of treatment.

Other platform trials have similarly investigated the role of biomarker-targeted treatments in G/GOJ cancers. The VIKTORY trial [106] aimed to classify advanced gastric cancer patients into eight subgroups and assign them to second-line trials based on molecular profiling results. With 772 patients recruited, around 15% of patients received biomarker-directed treatment and showed some encouraging survival outcomes compared to standard second-line treatment. The trial also showed that the targets and PD-L1 status changed over time [106].

The K-Umbrella Gastric Cancer Study [107] investigated druggable targets by IHC in HER2-negative gastric cancers and randomised patients to biomarker-guided or control group second-line treatment. The study found no statistically significant survival difference between biomarker-directed and standard-of-care treatments [107].

One of the reasons for some of these negative results could also be that tissue-based biomarker selection may have limited identification of biomarker-driven oncogenesis. For example, the observational SCRUM-Japan GOZILA study investigating ctDNA-based screening compared to tumour tissue sequencing showed that using ctDNA-based genotyping shortened the duration of screening, improved enrolment in the trial and did not compromise treatment efficacy compared to tissue-based testing [108]. This gives hope to future ctDNA-based trials adding further information about the heterogeneity of targets and improved precision medicine in G cancers.

### 8.2. Sequence and Combining Targeted Therapies in Gastric Cancer

The previously mentioned PANGEA trial [105] also built a prioritisation algorithm for patient samples where multiple targets are present. Showing that whilst the targets are often investigated separately, they are not mutually exclusive. This has especially become relevant now where recent developments in immunotherapy have made combinations of ICI and chemotherapy the new standard first-line treatment for a large subgroup of patients with gastric cancer.

However, for example, the recently positive SPOTLIGHT trial investigated CLDN18.2-targeted treatment compared to standard combination chemotherapy, and therefore for first-line gastric cancer it is not known which biomarker should be prioritised—CPS or CLDN18.2 positivity. In FGFR-targeted trials, the phase II FIGHT trial [4] similarly compared to standard combination chemotherapy, but ongoing phase III trials [87,109] are already using the ICI and chemotherapy backbone, and will yield more information about the mutual expression of FGFR alterations and PD-L1 scores. As previously mentioned, in HER2-positive cancers, combining HER2-targeted treatment and immunotherapy is currently under investigation in the KEYNOTE-811 trial with promising early results [55]. The biomarker used for inclusion in that trial was HER2+, but it is not known if all CPS levels will be responding equally to the quadruple combination, or if CPS cut-offs will be similarly predictive. At the moment, first-line immunotherapy and chemotherapy combinations are only licenced in the HER2-negative group, as Checkmate-649 excluded HER2+ tumours.

Therefore, we do not know how each of the current and in-development targeted treatments will actually fit into the treatment paradigm of advanced gastric cancers. Nevertheless, any improvement in the outcomes of these patients is very welcome as survival remains poor.

## 9. Conclusions

Whilst the first targeted treatment results for advanced gastric cancer were positive over a decade ago, the targets used in everyday practice for guiding treatment options remain minimal. With recent discoveries of new druggable targets in subgroups of patients, we are moving away from a one-size fits all approach of combination chemotherapy, but results from multiple platform and umbrella trials are still awaited to see how these extend survival outcomes. Due to aggressive disease biology, combination chemotherapy still remains an important backbone for gastric cancer. However, rapid improvements in immunotherapy have brought new hope for extending OS over the previously static one-year mark in this poor-prognosis patient group and, importantly, have given us new predictive biomarkers.

The evolving landscape of molecular alterations in gastric cancer and the multiple targets being assessed simultaneously will pose important questions about the sequencing of treatment, overlapping toxicity, necessary technologies required to perform these molecular tests and obviously resource implications of testing and precision medicine treatment in gastric cancer.

## Figures and Tables

**Figure 1 cancers-15-03248-f001:**
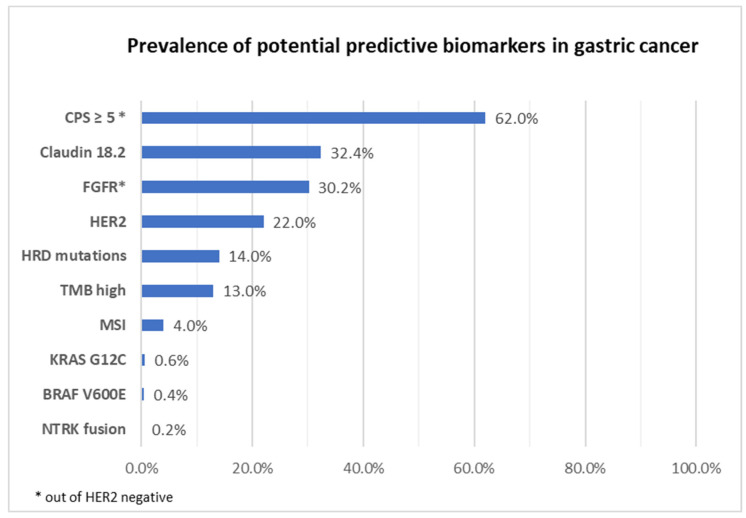
Prevalence of potential predictive biomarkers in advanced gastric cancer [2,3,4,5,6,7,8,9,10,11]. CPS—combine positive score; FGFR—fibroblast growth factor receptor; HER2—human epidermal growth factor receptor 2; HRD—homologous recombination deficiency; TMB—tumour mutational burden; MSI—microsatellite instable; KRAS—KRAS proto-oncogene; BRAF—BRAF-B-Raf proto-oncogene; NTRK—neurotrophic receptor tyrosine kinase.

## Data Availability

The data can be shared up on request.

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
