# Peer review of "Targeted Therapies and Developing Precision Medicine in Gastric Cancer"

_cancers, 2023, doi:10.3390/cancers15123248_

Round 1
Reviewer 1 Report
The article entitled “Targeted therapies and developing precision medicine in gastric cancer” by Rille Pihlak, et al. demonstrated that the authors present a lot of interests in this report. However, there are areas that need to be improved.
Minor comments
(1) How did authors obtain the data in Figure 1?
(2) Should authors list Phase 1 and 2 study in Table 1.
(3) Results of clinical trial in Table 1 are difficult to understand at a glance.
(4) In Table 1, is it correct that ATTRACTION-4 is described as phase 2 trial?
(5) Should authors list the results of many negative trials in Table 2.
(6) Describes in “3. HER2-targeted therapies” (Line 231-236, Line 253-269, Line 270-300, Line 307-313, Line 314-328, Line 329-351, Line 352-365), are somewhat redundant and needs shortening.
Author Response
- How did authors obtain the data in Figure 1?
References added to clarify where this data came from.
- Should authors list Phase 1 and 2 study in Table 1.
Large phase I-II trials were kept in Table 1, to illustrate results compared to other large trials.
- Results of clinical trial in Table 1 are difficult to understand at a glance.
Text in table harmonised, some less important data removed, like 95% confidence intervals, and text cleaned to read better.
- In Table 1, is it correct that ATTRACTION-4 is described as phase 2 trial?
Corrected that ATTRACTION-4 was a phase 2-3 trial.
- Should authors list the results of many negative trials in Table 2.
Many negative trials are in Table 2 to illustrate a selection of important trials in HER2+ disease. We tried to keep the table both short and relevant.
- Describes in “3. HER2-targeted therapies” (Line 231-236, Line 253-269, Line 270-300, Line 307-313, Line 314-328, Line 329-351, Line 352-365), are somewhat redundant and needs shortening.
This whole section has been reviewed and shortened.
Reviewer 2 Report
1. For MMR deficiency patients, phase 2 study of ipilimumab plus nivolumab is ongoing. Please cite the study (Cancers(Basel). 2021 Feb 15;13(4):805.doi: 10.3390/cancers13040805.).
2. In HER2 target section, Makiyama et al reported randomized phase 2 study of PTX plus trastuzumab vs PTX. It showed HER2 amplification decreased after trastuzumab containing regimen. Please cite the study(J clin Oncol 2020 Jun 10;38(17):1919-1927.doi: 10.1200/JCO.19.03077.).
Author Response
- For MMR deficiency patients, phase 2 study of ipilimumab plus nivolumab is ongoing. Please cite the study (Cancers(Basel). 2021 Feb 15;13(4):805.doi: 10.3390/cancers13040805.).
Added to the MMR deficient section in the manuscript.
- In HER2 target section, Makiyama et al reported randomized phase 2 study of PTX plus trastuzumab vs PTX. It showed HER2 amplification decreased after trastuzumab containing regimen. Please cite the study(J clin Oncol 2020 Jun 10;38(17):1919-1927.doi: 10.1200/JCO.19.03077.).
The study is already cited as reference nr 71, Table 2
Reviewer 3 Report
This is a well concepted and well written paper.
I suggest to add a visual abstract or a summarizing image.
Minor typos errors.
Author Response
Figure 1 and Table 1-2 are hopefully good at conveying the general message of the paper, therefore no further images were added, typos cleaned.
Text is currently in UK English, if need to, can change to US.